palaeontology/image processing/software

three-dimensional segmentation, *Drishti Paint*, computed tomography, 3D Freeform Painter, gradient threshold

**Author for correspondence:**
Jing Lu
e-mail: lujing@ivpp.ac.cn

# Three-dimensional segmentation of computed tomography data using *Drishti Paint*: new tools and developments

Yuzhi Hu[1,2], Ajay Limaye[3] and Jing Lu[4]

[1]Department of Applied Mathematics, Research School of Physics, Australian National University, Canberra, ACT 2601, Australia
[2]Research School of Earth Sciences, Australian National University, Canberra, ACT 2601, Australia
[3]National Computational Infrastructure, Building 143, Corner of Ward Road and Garran Road, Ward Rd, Canberra, ACT 2601, Australia
[4]Institute of Vertebrate Paleontology and Paleoanthropology, Chinese Academy of Sciences, Beijing 100044, People's Republic of China

YH, 0000-0002-4794-0935; JL, 0000-0002-5791-4749

Computed tomography (CT) has become very widely used in scientific and medical research and industry for its non-destructive and high-resolution means of detecting internal structure. Three-dimensional segmentation of computed tomography data sheds light on internal features of target objects. Three-dimensional segmentation of CT data is supported by various well-established software programs, but the powerful functionalities and capabilities of open-source software have not been fully revealed. Here, we present a new release of the open-source volume exploration, rendering and three-dimensional segmentation software, *Drishti* v. 2.7. We introduce a new tool for thresholding volume data (i.e. gradient thresholding) and a protocol for performing three-dimensional segmentation using the 3D Freeform Painter tool. These new tools and workflow enable more accurate and precise digital reconstruction, three-dimensional modelling and three-dimensional printing results. We use scan data of a fossil fish as a case study, but our procedure is widely applicable in biological, medical and industrial research.

## 1. Introduction

Computed tomography (CT) has become widely used in medical, scientific and industrial research [1–5]. To have a better understanding of the data generated from CT, micro-CT and

synchrotron radiation phase-contrast imaging, three-dimensional scientific visualization has become more and more crucial for researchers to obtain better insights [6–9]. For over three decades, the field of three-dimensional scientific visualization has been developed considerably with many new techniques to visualize and analyse data more accurately [10–19]. Two techniques have emerged to provide different visualizations: surface rendering, the method of interpreting datasets by generating a set of polygons that represent surfaces of the desired feature; and volume rendering [10], for the reconstruction of three-dimensional structures both internally and externally. Volume rendering represents three-dimensional objects as a collection of cube-like building blocks called voxels, or volume elements. A range of well-established commercial software, such as Mimics, VG Studio and AVIZO provide numerous functionalities and good rendering outputs. However, the potential of open-source software, which is both freely available and easy to access, has not been fully explored. Three-dimensional segmentation, segmenting the internal region of interest in sequences of images, is a vital tool for investigating and understanding the internal structures of target objects [6,20–22]. Two-dimensional segmentation uses each image or slice in a volumetric dataset to construct a three-dimensional presentation. By contrast, three-dimensional segmenting methods, the main focus of this paper, use thresholding, edge detection, clustering or region growing techniques to group pixels based on brightness, colour or texture [21], and then render them as discrete objects.

Among 10 well-known cross-platform software programs [23], five use both surface and volume rendering, but only *Drishti* [24] with 'an intuitive user interface' [23], can perform three-dimensional segmentation directly on a volume (using the new tool presented here, 3D Freeform Painter). *Drishti* uses direct volume rendering with voxel ray casting and texture slicing algorithms; combinations of local and global illuminations along with a two-dimensional transfer function which merges colour and opacity to volume to provide realism on different materials and textures through lighting, shading or shadowing via different user-generated light volumes and change of opacity. As an open-source volume exploration and rendering software [24], *Drishti* has demonstrated three-dimensional rendering of high quality [6,25–27].

Here, we present the most recent release of *Drishti* v. 2.7, and introduce new tools for three-dimensional segmentation (two- and three-dimensional painters) of volumetric data. We also introduce a new tool-gradient threshold and suggest protocols for how to perform three-dimensional segmentation using *Drishti Paint* v. 2.7 efficiently and precisely, using the CT scan data of a fossil fish as a case study. New features in *Drishti*, such as mesh generation and simplification, are also explained and discussed.

Volume rendering does require datasets to be loaded into graphics memory. The minimum requirements to run *Drishti* are graphics processing unit (GPU)—a graphics card with 1 GB memory— and random access memory (RAM)—4GB. *Drishti* does not have a specific requirement for the central processing unit (CPU). For readers who do not have much computing power, we suggest trying other open-source software such as SPIERS [28] and 3D Slicer [29,30]. Both software perform excellent surface rendering and allow meshes to be handled, which can then be modified in Meshlab [31] or Blender [32].

# 2. Materials and methods

## 2.1. Materials

The CT data used for this study is from a 400-million-year-old fossil fish from Burrinjuck, near Canberra, southeastern Australia (ANU V244, held at the Department of Applied Mathematics, Research School of Physics, Australian National University, Canberra). The whole specimen was scanned in 2011 and rescanned in 2015 at CT Lab, ANU [33]. The voxel size of the second scan is 21 µm.

## 2.2. Methods

CT data were reconstructed using an in-house software called *Mango* (https://physics.anu.edu.au/ appmaths/capabilities/mango.php). *Mango* can perform three-dimensional tomographic reconstruction on CPU or GPU clusters. The CPU-only code is limited to analytic methods: Feldkamp–Davis–Kress (FDK) filtered back-projection (FBP) for a circular X-ray source trajectory and Katsevich FBP for helical/double-helical X-ray source trajectories [34]. The GPU code can also perform multi-grid iterative reconstruction [35] for a space-filling X-ray source trajectory [1]. The reconstruction code has software for automatic geometric alignment capabilities [36], X-ray source movement correction [37] and component or rigid-body sample movement correction [38].

The original whole specimen CT dataset was saved as 16-bit images. These data were loaded into *Drishti Import* v. 2.7. The contrast was incremented with the help of histograms, and slices were filtered by selecting the best range on the tomogram. The raw data of the whole specimen was then cropped to focus on a

R. Soc. Open Sci. **7**: 201033

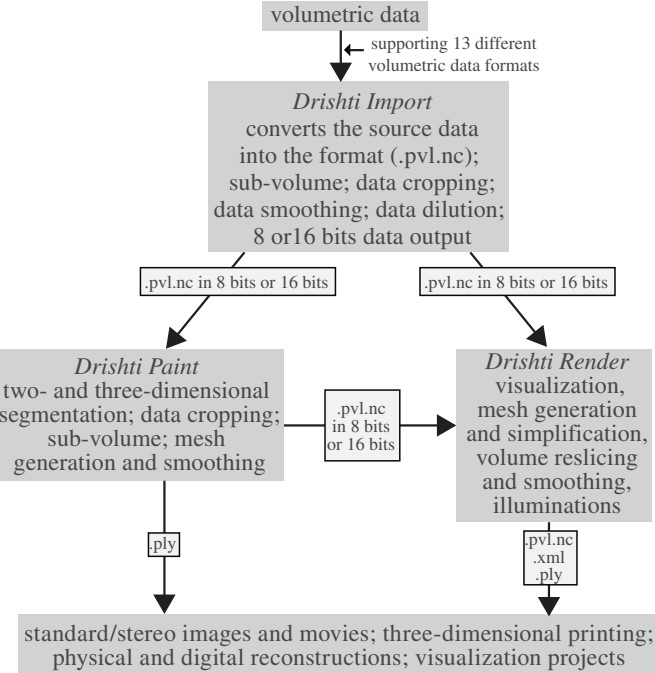

**Figure 1.** The general workflow for operating *Drishti* v. 2.7. Data formats in this figure are given as abbreviations, for which detailed information is presented in the electronic supplementary material, table S1.

selected region for the right cheek complex, which was segmented out and is used as a case study here. Raw data of the selected region of interest was saved in the format *.pvl.nc (i.e. processed volume format) and can be downloaded from figshare: https://doi.org/10.6084/m9.figshare.12073809. We developed a general protocol to segment the three-dimensional volumetric data using *Drishti Paint* v. 2.7 (see electronic supplementary material for the detailed segmentation procedure).

# 3. Results and discussion

Here, we highlight the workflow and key features of the *Paint* module for *Drishti* and the general applications of *Drishti*. We have developed a protocol and a workflow, summarized in figure 1, to facilitate a more efficient and accessible (i.e. open-source freeware) means of overcoming the time-intensive nature of preparing fossils digitally and extracting complex information from samples. Detailed information on new tools-gradient thresholding and 3D Freeform Painter are provided in the electronic supplementary material. Our protocol and workflow (figure 1) can be used as a baseline or starting point to be adapted to obtain an ideal result for any given dataset. The electronic supplementary material also provides details on other aspects of *Drishti*, including installation instructions, a summary of all *Drishti*-supported import formats, additional learning resources and other background information.

Three modules, *Drishti Import*, *Drishti Render* and *Drishti Paint*, have different capacities and features, which can be combined to ensure an accurate and precise segmentation along with volume rendering to help visualize a region-of-interest for detailed analysis. *Drishti Paint* uses a variety of discontinuity detection-based and similarity detection-based image segmentation approaches. These two approaches are both implemented in two modes in *Drishti Paint* v. 2.7, i.e. 'Graph Cut' and 'Curve' (previously known as livewire).

The right cheek complex has been segmented from the original CT dataset for our case study. Figure 2 shows an overview of our case study, where a fossil fish skull (figure 2*a,b*) is cropped (figure 2*c*) and segmented using the latest tools in *Drishti* v. 2.7 (figure 2*d–e*). Processed data is exported as a separate volume by using the tagging function to extract the region of interest. Segmentation (in this cheek complex example) was carried out in 16-bits full resolution in alignment with the raw data to include all information from the original scan (8-bits resolution could be used, if computing power is limited).

Two transformations in mathematical morphology (i.e. dilation and erosion) can be implemented in *Drishti Paint*. Dilation adds pixels to the boundaries of objects in an image, while erosion removes pixels on object boundaries. Morphological dilation makes objects more visible and fills in small holes in objects. Morphological erosion removes islands and small objects so that only substantive objects remain. These two transformations can be used in any order. We used a combination of these two

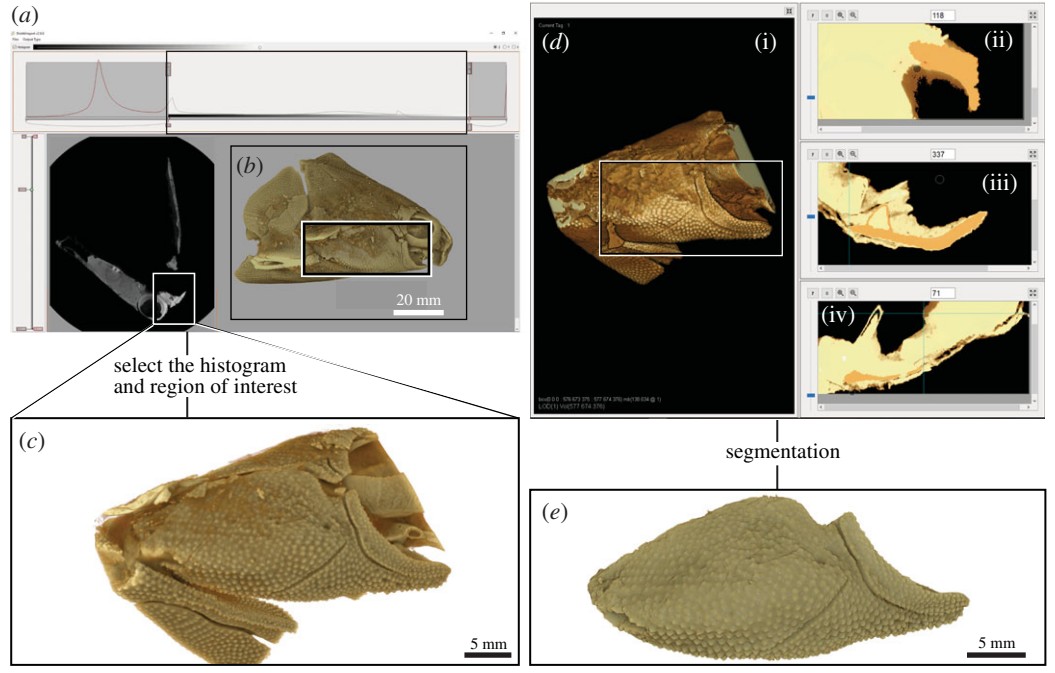

**Figure 2.** Extracting a three-dimensional virtual volumetric model in *Drishti* v. 2.7. (*a*) original data of the whole fossil fish specimen (V244) used for our case study. (*b*) original data in *Drishti Render* v. 2.7 (right lateral view of skull) showing the selected region of interest (*c*). (*d*) Region of interest in *Drishti Paint* v. 2.7. (*e*) Extracted right cheek complex in *Drishti Render* v. 2.7 (external view).

transformations with Graph Cut and 3D Freeform Painter. These two transformations are more useful when using clean and high-contrast data as there is a risk of changing the morphology of the input datasets by doing multiple transformations. We recommend keeping a copy of the original volume data, before doing multiple transformations, to avoid losing any information of the original dataset. Interpolation is another tool which can be used in Curve mode. However, interpolation is not recommended for palaeontological datasets (due to lack of reproducibility).

Three types of gradient thresholding are developed and implemented in *Drishti Paint* v. 2.7. Values thresholding was developed in *Drishti* v. 2.3.1, but this is the first implementation of gradient thresholding in an open-source volume rendering software program. Gradient thresholding can be combined with values thresholding to clarify and more precisely identify the boundaries between different phases, which then makes the three-dimensional segmentation process easier. Multiple-thresholding (i.e. use both values and gradient thresholding) in *Drishti Paint* v. 2.7 is beneficial for volume segmentation and usually is the first step towards segmenting a volume (figure 2*d*).

In the gradient thresholding tool, gradient type 1 uses the magnitude of central difference (i.e. surface gradient vector). Gradient type 2 takes the sum of all voxel values in one neighbourhood (i.e. $3 \times 3 \times 3$ box), then subtracts the central value from the above sum and divides it by 10. The absolute of the difference between the central value and the above result after dividing by 10 is restricted to a value between 0.0 and 1.0. Gradient type 3 takes the sum of all voxel values in two neighbourhoods (i.e. $5 \times 5 \times 5$ box), then subtracts the central value from the above sum and divides it by 70. As for type 2, the absolute of this difference is also restricted to a value between 0.0 and 1.0. In gradient types 2 and 3, the 'gradient' value is shifted according to the voxel value (i.e. gradients for lower voxel values will be smaller compared to those for higher voxel values). For our case study, type 2 gradient thresholding combined with values thresholding were used to select the range of the histogram for segmenting out the right cheek complex. Gradient thresholding is also sensitive to noise and intensity inhomogeneities. There is no best combination when it comes to which gradient type one should use. Generally speaking, most boundaries can be clarified using gradient types 2 and 3. By contrast, gradient type 1 is more applicable with a cleaner dataset with fewer phases, such as medical datasets.

The segmented right cheek unit has also been extracted as surface meshes (figure 3), which also illustrates the use of the new mesh simplification function in *Drishti* v. 2.7. In this case, the inner surface of the cheek unit is used to test different simplification procedures, because of its more complex structure (the upper jaw cartilage of the fish has surface ossification and is fused to the inside of the cheek bones). The surface mesh data went through different stages of simplification (see electronic supplementary

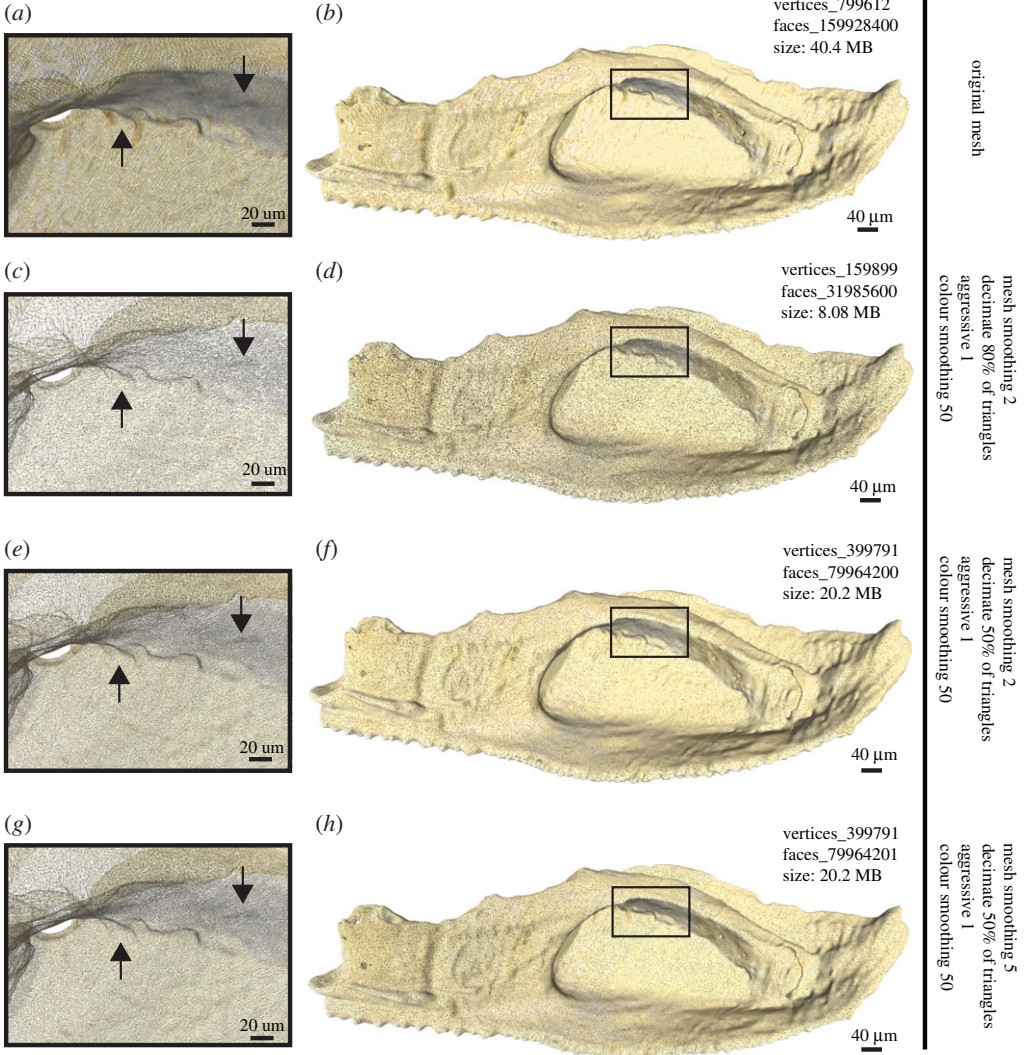

**Figure 3.** Three-dimensional surface meshes of the extracted right cheek unit of figure 2e, showing the inner surface with attached perichondrally ossified jaw cartilages. The differences of vertex count and file size after different mesh simplification applications are summarized for each. The vertex count was calculated in MeshLab v. 1.3.4 [31]. Arrows indicate the reference points for detailed structure compared under the different simplified procedures (see discussion in the text).

material) to produce good quality surface models with relatively small file size. This allows faster physical analysing (e.g. finite-element analysis) and three-dimensional printing. Different mesh simplification gives different number of vertices, which affects the presentation of the information and the detection of details (figure 3a,c,e,g). When more triangles have been decimated, the loss of information results in a reduction in file size, which allows for easier sharing of the digital three-dimensional model [20], or uploading the model to an open data repository for educating the public [21]. By comparing the number of details lost or preserved in our case study (arrows in figure 3), a mesh smoothing factor of 2 and 50% decimation is recommended (figure 3e,f). This halves the data size but still preserves most of the information. Three-dimensional models can then be used to generate three-dimensional printouts [39] or three-dimensional portable documents [40], which can test a previous hypothesis, help with functional morphology investigations, or be used for science communication and public outreach [41].

## 4. Conclusion

The new tools and proposed workflow to segment volumetric data in the latest *Drishti* version (*Drishti* v. 2.7) are generally accessible to the public, to provide more accurate and precise digital reconstruction, three-dimensional modelling and three-dimensional printing. The protocol and workflow we have developed

can be used as a framework to segment computed tomography data and other forms of volumetric data and is widely applicable in biological, medical and earth science research. Our work will facilitate further cross-discipline collaborations using three-dimensional segmentation of computed tomography data.

Data accessibility. Three-dimensional surface mesh data of the segmented right cheek complex with different levels of simplification and the cropped raw data of the region-of-interest including the right cheek complex are available from the figshare Repository: https://doi.org/10.6084/m9.figshare.12073809.
*Drishti* v. 2.7 (Windows and Linux versions) and relevant code for this research work are stored in GitHub: https://github.com/nci/drishti and have been archived within the Zenodo repository: https://doi.org/10.5281/zenodo.4092471.

Authors' contributions. J.L. and Y.H. designed the study. Y.H., J.L. and A.L. performed the research and drafted the manuscript. Y.H. and J.L. prepared the figures. A.L. developed *Drishti* v. 2.7. All authors revised the manuscript. Y.H. and A.L. contributed equally.

Competing interests. The authors declare no competing interests.

Funding. This research was funded by the Strategic Priority Research Program of the Chinese Academy of Sciences (grant no. XDB26000000) and the National Natural Science Foundation of China (grant no. 41872023). Y.H. was supported by a Postgraduate Research Scholarship at the Research School of Physics, Australian National University. The development of *Drishti* is supported by National Computational Infrastructure, Australian National University. CT scans and three-dimensional printing are supported by the Department of Applied Mathematics, Research School of Physics and ANU CT Lab, with funding support from Prof. T. Senden and Australian Research Council Discovery Grant DP160102460.

Acknowledgements. We thank Prof. T. Senden (Director, RSPhys, ANU) for provision of facilities, CT scanning and general support for palaeobiological research at ANU. We thank Dr G. Young for help with proofreading; Dr M. Turner for CT scanning; Dr L. Beeching for laboratory support and Dr A. Kingston for help with *Mango*. We thank Dr Russell Garwood for his involvement towards testing the Linux version of *Drishti* v. 2.7. We thank Dr Garwood and another anonymous reviewer for providing valuable suggestions that greatly improved this paper.

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
