## [Reviewer comments · Royal Society Open Science]

Review History

RSOS-201033.R0 (Original submission)

Review form: Reviewer 1 (Jérémy Tissier)

Is the manuscript scientifically sound in its present form?

Yes

Are the interpretations and conclusions justified by the results?

Yes

Is the language acceptable?

No

Do you have any ethical concerns with this paper?

No

Have you any concerns about statistical analyses in this paper?

No

Recommendation?

Accept with minor revision (please list in comments)

Comments to the Author(s)

This manuscript presents a new version of the Drishti software, and a new tool for segmentation: Drishti Paint. I very much appreciate the fact that this software is open source, as it is always good to have free alternatives. However, I believe it would have been even more interesting to present and discuss the tools and features offered by this software for the segmentation process, which are not presented in details.

Actually, the segmentation process is only very vaguely discussed ("Two transformations in mathematical morphology (i.e. erosion and dilation) were implemented in Drishti Paint and we used a combination of these two transformations with 3D Freeform Painter to help with a faster segmentation process." 1.107-109), and the data do not permit replication of the results because the raw 3D data are not provided. Thus the segmentation cannot be replicated.

Thus, I suggest to carefully check the English before publication, to provide the raw 3D data used for the segmentation, and to discuss further the detailed segmentation procedure and the Drishti features used.

Thus I recommend publication of this manuscript only after minor revision.

Review form: Reviewer 2 (Tom Challands)

Is the manuscript scientifically sound in its present form?

No

Are the interpretations and conclusions justified by the results?

No

Is the language acceptable?

No

Do you have any ethical concerns with this paper?

No

Have you any concerns about statistical analyses in this paper?

No

Recommendation?

Major revision is needed (please make suggestions in comments)

Comments to the Author(s)

This manuscript outlines briefly how the authors have used a well-known open-access piece of software for rendering and segmenting microCT and other similar datatypes. The version of the software they use is actually now outdated as v2.7 has been released and the information they provide is not really any different from that which is present on the github site for the software i.e. the new features in the version they use (v2.6.6). Indeed, the github site and accompanying YouTube tutorials provide a more detailed framework and protocol for usage of Drishti than has been presented here. This is a shame as a definitive citable manuscript detailing how to use Drishti would be extremely useful to the public.

As it stands I cannot recommend this paper for publication. It requires some major changes in terms of grammar, detail and data accessibility.

A protocol would describe in detail the procedure used in Drishti Paint to segment the elements from the original scan data. The original scan data is the raw data that should be included for users to replicate the results from, not the resulting segmented pvl.nc file. I cannot segment out something that has already been segmented.

I understand that the authors may not want to release a microCT dataset of a specimen that has not been described yet but if that is so they should use a microCT dataset that they are willing to make freely available. They could even download one from elsewhere that has been published.

Furthermore, the authors do not describe anything new from that which is freely available in the instructions and tutorials on the Drishti github page. In fact, the material available there is far more informative in terms of segmentation protocols.

I would consider reviewing this manuscript again if the following changes are made but if these steps are not done I cannot recommend it for publication in Royal Society Open Science, a journal that promotes data sharing and open access:

1 - Make the entire microCT dataset available. If the authors do not wish to make this dataset available then use a different one. If they do not have a different dataset to use then they could crop the original dataset to around the region of interest to provide enough data that the user could segment the cheek and tooth but not be able to access all the other elements. This would solve the risk of another party describing the specimen before the authors do. It is the process this manuscript purports to describe that is important as this manuscript describes a method and not a specimen.

From the website of Royal Society Open Science, one of the distinguishing features of the journal is that "articles embody open data principles". Unless the original microCT scan data is released this manuscript does not adhere to the principles of the journal.

2 - Describe in detail each step that a user must do to in Drishti Paint to segment the specific element used in this example. Statements such as line 110-111 "The right cheek complex has been segmented from the original CT dataset of the whole specimen (figure 2)." does not tell me how to do this. These instructions could be included in the supplementary info.

I have not been able to do this because the data has not been made available and the individual steps have not been described.

In the case of the segmented placoderm tooth with the internal canals, the authors may have achieved better results using the 'tubes' function in Drishti Paint. Perhaps they did use this function or perhaps they used the gradient function, they do not say how they segmented the tubes. In fact, how do you access the tubes function in Drishti as shown in the online tutorial video?!? Spacebar to access the command help menu does not function in v2.6.6 or 2.7 in the windows zip files from github.

3 - Provide detailed compilation and installation instructions for linux and mac. The authors provided a further brief set of steps on how to install Drishti for linux following my request for information on this matter. It did not work. I understand that linux distributions are all different and an individual's linux set up unlikely matches that of another but I was still unable to compile and install it after editing the .pri file for my setup and then subsequently altering the makefile produced by QTCreator to account for the versions of the dependencies that the authors provided me. Both systems are unstable and crash frequently in Wine (windows compatibility layer for linux operating systems).

4 - Correct English grammar. Despite native English speakers being acknowledged for their assistance in proof-reading the English grammar is still not acceptable for publication.

Other detailed points

Title – Drishti Paint is not a new tool and has been present in many versions of Drishti

Line 33 Should you include microCT and synchrotron data here as well as Drishti can handle all these. Yes, they are all essentially CT but perhaps not everyone knows that.

Lines 36-38 “For the last four decades, the field of 3D scientific visualization field has have been developed considerably with many new techniques available ways to visualize and analyse data more accurately [10-19].”

Line 44. By which community? All communities, the palaeo, biological, medical communities?

Line 46. 3D segmentation is not necessarily about segmenting structure but regions of interest. I would use ‘regions of interest’ rather than structures. It depends what type of data you are working with.

Lines 48 -50. You omit manual segmentation involving drawing the shape of each region of interest and, sometimes, using interpolation. Though perhaps the most commonly used in palaeo datasets it is also the most fraught with uncertainty, error and lack of reproducibility. This should be mentioned somewhere in the manuscript as you are presenting a workflow that is reproducible.

Lines 50–52. This is not great English I’m afraid. I would reword along the lines of: “Compared with researches carried out in the image processing field, more requirements have to be met towards building and Despite this researchers in many fields still require the development of developing applications and techniques which are require the use of more efficient and effective techniques at segmenting to save research time and resources.”

Line 54. “... high quality 3D rendering” better than “... superb visualisation outcomes.”. Other rendering packages are good as well so it might be worth explaining what gives Drishti rendering its unique quality. Ajay should be able to add this in.

Line 55. Drishti 2.7 is now available. I assume that the same Drishti paint tools are present in 2.7 in which case the latest version should be cited.

Lines 80-84. This is the crux (most important part) of the manuscript and it is treated very lightly. In the abstract the authors state they introduce a “...protocol for performing 3D segmentation” but they have not. They have simply stated that the data was segmented using Drishti Paint.

Review form: Reviewer 3 (Russell Garwood)

Is the manuscript scientifically sound in its present form?

Yes

Are the interpretations and conclusions justified by the results?

Yes

Is the language acceptable?

Yes

Do you have any ethical concerns with this paper?

No

Have you any concerns about statistical analyses in this paper?

No

Recommendation?

Accept with minor revision (please list in comments)

Comments to the Author(s)

Please see attachment (Appendix A).

Decision letter (RSOS-201033.R0)

Dear Dr Lu

The Editors assigned to your paper RSOS-201033 "A new tool for 3D segmentation of computed tomography data: Drishti Paint and its applications" have now received comments from reviewers and would like you to revise the paper in accordance with the reviewer comments and any comments from the Editors. Please note this decision does not guarantee eventual acceptance.

You will see that all three reviewers see the paper potentially as a valuable contribution, but all recommend corrections, improvements and clarifications before publication and Referee 2 in particular has recommended that your paper requires major revision in order to be suitable for publication -- therefore please pay particular attention to the comments of Referee 2.

Please submit your revised manuscript and required files (see below) no later than 21 days from today's (ie 27-Aug-2020) date. Note: the ScholarOne system will 'lock' if submission of the revision is attempted 21 or more days after the deadline. If you do not think you will be able to meet this deadline please contact the editorial office immediately.

on behalf of Prof Peter Haynes (Subject Editor)
openscience@royalsociety.org

Associate Editor Comments to Author :

Thank you for your submission. The reviewers are generally in favour of publication of the paper; however, a few modifications are recommended before further consideration is possible, and the revised paper may be returned to the reviewers for a final assessment.

Reviewer comments to Author:

Reviewer: 1

Comments to the Author(s)

This manuscript presents a new version of the Drishti software, and a new tool for segmentation: Drishti Paint. I very much appreciate the fact that this software is open source, as it is always good to have free alternatives. However, I believe it would have been even more interesting to present and discuss the tools and features offered by this software for the segmentation process, which are not presented in details.

Actually, the segmentation process is only very vaguely discussed ("Two transformations in mathematical morphology (i.e. erosion and dilation) were implemented in Drishti Paint and we used a combination of these two transformations with 3D Freeform Painter to help with a faster segmentation process." 1.107-109), and the data do not permit replication of the results because the raw 3D data are not provided. Thus the segmentation cannot be replicated.

Thus, I suggest to carefully check the English before publication, to provide the raw 3D data used for the segmentation, and to discuss further the detailed segmentation procedure and the Drishti features used.

Thus I recommend publication of this manuscript only after minor revision.

Reviewer: 2

Comments to the Author(s)

This manuscript outlines briefly how the authors have used a well-known open-access piece of software for rendering and segmenting microCT and other similar datatypes. The version of the software they use is actually now outdated as v2.7 has been released and the information they provide is not really any different from that which is present on the github site for the software i.e. the new features in the version they use (v2.6.6). Indeed, the github site and accompanying YouTube tutorials provide a more detailed framework and protocol for usage of Drishti than has been presented here. This is a shame as a definitive citable manuscript detailing how to use Drishti would be extremely useful to the public.

As it stands I cannot recommend this paper for publication. It requires some major changes in terms of grammar, detail and data accessibility.

A protocol would describe in detail the procedure used in Drishti Paint to segment the elements from the original scan data. The original scan data is the raw data that should be included for users to replicate the results from, not the resulting segmented pvl.nc file. I cannot segment out something that has already been segmented.

I understand that the authors may not want to release a microCT dataset of a specimen that has not been described yet but if that is so they should use a microCT dataset that they are willing to make freely available. They could even download one from elsewhere that has been published.

Furthermore, the authors do not describe anything new from that which is freely available in the instructions and tutorials on the Drishti github page. In fact, the material available there is far more informative in terms of segmentation protocols.

I would consider reviewing this manuscript again if the following changes are made but if these steps are not done I cannot recommend it for publication in Royal Society Open Science, a journal that promotes data sharing and open access:

1 - Make the entire microCT dataset available. If the authors do not wish to make this dataset available then use a different one. If they do not have a different dataset to use then they could crop the original dataset to around the region of interest to provide enough data that the user could segment the cheek and tooth but not be able to access all the other elements. This would solve the risk of another party describing the specimen before the authors do. It is the process this manuscript purports to describe that is important as this manuscript describes a method and not a specimen.

From the website of Royal Society Open Science, one of the distinguishing features of the journal is that "articles embody open data principles". Unless the original microCT scan data is released this manuscript does not adhere to the principles of the journal.

2 - Describe in detail each step that a user must do to in Drishti Paint to segment the specific element used in this example. Statements such as line 110-111 "The right cheek complex has been segmented from the original CT dataset of the whole specimen (figure 2)." does not tell me how to do this. These instructions could be included in the supplementary info.

I have not been able to do this because the data has not been made available and the individual steps have not been described.

In the case of the segmented placoderm tooth with the internal canals, the authors may have achieved better results using the 'tubes' function in Drishti Paint. Perhaps they did use this function or perhaps they used the gradient function, they do not say how they segmented the tubes. In fact, how do you access the tubes function in Drishti as shown in the online tutorial video?!? Spacebar to access the command help menu does not function in v2.6.6 or 2.7 in the windows zip files from github.

3 - Provide detailed compilation and installation instructions for linux and mac. The authors provided a further brief set of steps on how to install Drishti for linux following my request for information on this matter. It did not work. I understand that linux distributions are all different and an individual's linux set up unlikely matches that of another but I was still unable to compile and install it after editing the .pri file for my setup and then subsequently altering the makefile produced by QTCreator to account for the versions of the dependencies that the authors provided me. Both systems are unstable and crash frequently in Wine (windows compatibility layer for linux operating systems).

4 - Correct English grammar. Despite native English speakers being acknowledged for their assistance in proof-reading the English grammar is still not acceptable for publication.

Other detailed points

Title – Drishti Paint is not a new tool and has been present in many versions of Drishti

Line 33 Should you include microCT and synchrotron data here as well as Drishti can handle all these. Yes, they are all essentially CT but perhaps not everyone knows that.

Lines 36-38 “For the last four decades, the field of 3D scientific visualization field has have been developed considerably with many new techniques available ways to visualize and analyse data more accurately [10-19].”

Line 44. By which community? All communities, the palaeo, biological, medical communities?

Line 46. 3D segmentation is not necessarily about segmenting structure but regions of interest. I would use ‘regions of interest’ rather than structures. It depends what type of data you are working with.

Lines 48 -50. You omit manual segmentation involving drawing the shape of each region of interest and, sometimes, using interpolation. Though perhaps the most commonly used in palaeo datasets it is also the most fraught with uncertainty, error and lack of reproducibility. This should be mentioned somewhere in the manuscript as you are presenting a workflow that is reproducible.

Lines 50-52. This is not great English I’m afraid. I would reword along the lines of: “Compared with researches carried out in the image processing field, more requirements have to be met towards building and Despite this researchers in many fields still require the development of developing applications and techniques which are require the use of more efficient and effective techniques at segmenting to save research time and resources.”

Line 54. “... high quality 3D rendering” better than “... superb visualisation outcomes.”. Other rendering packages are good as well so it might be worth explaining what gives Drishti rendering its unique quality. Ajay should be able to add this in.

Line 55. Drishti 2.7 is now available. I assume that the same Drishti paint tools are present in 2.7 in which case the latest version should be cited.

Lines 80-84. This is the crux (most important part) of the manuscript and it is treated very lightly. In the abstract the authors state they introduce a “...protocol for performing 3D segmentation” but they have not. They have simply stated that the data was segmented using Drishti Paint.

Reviewer: 3

Comments to the Author(s)

Please see attachment

===PREPARING YOUR MANUSCRIPT===

- one version identifying all the changes that have been made (for instance, in coloured highlight, in bold text, or tracked changes);
- a 'clean' version of the new manuscript that incorporates the changes made, but does not highlight them. This version will be used for typesetting if your manuscript is accepted.

===PREPARING YOUR REVISION IN SCHOLARONE===

- Ensure that your data access statement meets the requirements at <https://royalsociety.org/journals/authors/author-guidelines/#data>. You should ensure that you cite the dataset in your reference list. If you have deposited data etc in the Dryad repository, please include both the 'For publication' link and 'For review' link at this stage.
- If you are requesting an article processing charge waiver, you must select the relevant waiver option (if requesting a discretionary waiver, the form should have been uploaded at Step 3 'File upload' above).
- If you have uploaded ESM files, please ensure you follow the guidance at <https://royalsociety.org/journals/authors/author-guidelines/#supplementary-material> to include a suitable title and informative caption. An example of appropriate titling and captioning may be found at https://figshare.com/articles/Table_S2_from_Is_there_a_trade-off_between_peak_performance_and_performance_breadth_across_temperatures_for_aerobic_scope_in_teleost_fishes_/3843624.

Author's Response to Decision Letter for (RSOS-201033.R0)

See Appendix B.

RSOS-201033.R1 (Revision)

Review form: Reviewer 1 (Jérémy Tissier)

Is the manuscript scientifically sound in its present form?

Yes

Are the interpretations and conclusions justified by the results?

Yes

Is the language acceptable?

Yes

Do you have any ethical concerns with this paper?

No

Have you any concerns about statistical analyses in this paper?

No

Recommendation?

Accept as is

Comments to the Author(s)

Dear authors,

Thank you for taking into accounts all of the remarks that were suggested on the first review. The new manuscript is now very clear, and should be very helpful to the readers. The supplementary material, in particular, is very complete and easy to follow. I was able to replicace some of the results. All the data, 3D models and raw data are now provided.

I think the manuscript is now suitable for publication.

Regards.

Review form: Reviewer 2 (Tom Challands)

Is the manuscript scientifically sound in its present form?

Yes

Are the interpretations and conclusions justified by the results?

Yes

Is the language acceptable?

Yes

Do you have any ethical concerns with this paper?

No

Have you any concerns about statistical analyses in this paper?

No

Recommendation?

Accept with minor revision (please list in comments)

Comments to the Author(s)

This revision (a very thorough one!) provides a comprehensive introduction to the new segmentation tools in Drishti 2.6.7. The changes that the authors have made now provide a reproducible workflow with accessible data with greatly improves the citability of this contribution.

With the few typographical and grammatical changes made as outlined below I would recommend this article for publication. In the mean time I look forward to the linux release of Drishti 2.6.7.

Decision letter (RSOS-201033.R1)

Dear Dr Lu

On behalf of the Editors, we are pleased to inform you that your Manuscript RSOS-201033.R1 "3D segmentation of computed tomography data using Drishti Paint: new tools and developments" has been accepted for publication in Royal Society Open Science subject to minor revision in accordance with the referees' reports. Please find the referees' comments along with any feedback

from the Editors below my signature. (Reviewer 2 in particular has provided a small number of detailed comments that we expect you will wish to address.)

Please submit your revised manuscript and required files (see below) no later than 7 days from today's (ie 18-Nov-2020) date. Note: the ScholarOne system will 'lock' if submission of the revision is attempted 7 or more days after the deadline. If you do not think you will be able to meet this deadline please contact the editorial office immediately.

on behalf of Prof Peter Haynes (Subject Editor)
openscience@royalsociety.org

Associate Editor Comments to Author:

Your paper is essentially ready for acceptance; however, as one of the reviewers has a few typographical recommendations, we'd suggest you incorporate those changes before the paper is accepted.

Reviewer comments to Author:

Reviewer: 2

Comments to the Author(s)

This revision (a very thorough one!) provides a comprehensive introduction to the new segmentation tools in Drishti 2.6.7. The changes that the authors have made now provide a reproducible workflow with accessible data with greatly improves the citability of this contribution.

With the few typographical and grammatical changes made as outlined below I would recommend this article for publication. In the mean time I look forward to the linux release of Drishti 2.6.7.

Reviewer: 1

Comments to the Author(s)

Dear authors,

Thank you for taking into accounts all of the remarks that were suggested on the first review. The new manuscript is now very clear, and should be very helpful to the readers. The supplementary

material, in particular, is very complete and easy to follow. I was able to replicate some of the results. All the data, 3D models and raw data are now provided.

I think the manuscript is now suitable for publication.

Regards.

===PREPARING YOUR MANUSCRIPT===

- one version identifying all the changes that have been made (for instance, in coloured highlight, in bold text, or tracked changes);
- a 'clean' version of the new manuscript that incorporates the changes made, but does not highlight them. This version will be used for typesetting.

===PREPARING YOUR REVISION IN SCHOLARONE===

- 1) One version identifying all the changes that have been made (for instance, in coloured highlight, in bold text, or tracked changes);
 - 2) A 'clean' version of the new manuscript that incorporates the changes made, but does not highlight them.
 - An individual file of each figure (EPS or print-quality PDF preferred [either format should be produced directly from original creation package], or original software format).
 - An editable file of each table (.doc, .docx, .xls, .xlsx, or .csv).
 - An editable file of all figure and table captions.
- Note: you may upload the figure, table, and caption files in a single Zip folder.
- Any electronic supplementary material (ESM).
 - If you are requesting a discretionary waiver for the article processing charge, the waiver form must be included at this step.
 - If you are providing image files for potential cover images, please upload these at this step, and inform the editorial office you have done so. You must hold the copyright to any image provided.
 - A copy of your point-by-point response to referees and Editors. This will expedite the preparation of your proof.

- Ensure that your data access statement meets the requirements at <https://royalsociety.org/journals/authors/author-guidelines/#data>. You should ensure that you cite the dataset in your reference list. If you have deposited data etc in the Dryad repository, please only include the 'For publication' link at this stage. You should remove the 'For review' link.
- If you are requesting an article processing charge waiver, you must select the relevant waiver option (if requesting a discretionary waiver, the form should have been uploaded at Step 3 'File upload' above).
- If you have uploaded ESM files, please ensure you follow the guidance at <https://royalsociety.org/journals/authors/author-guidelines/#supplementary-material> to include a suitable title and informative caption. An example of appropriate titling and captioning may be found at [https://figshare.com/articles/Table_S2_from_Is_there_a_trade-off_between_peak_performance_and_performance_breadth_across_temperatures_for_aerobic_sc ope_in_teleost_fishes_/3843624](https://figshare.com/articles/Table_S2_from_Is_there_a_trade-off_between_peak_performance_and_performance_breadth_across_temperatures_for_aerobic_scope_in_teleost_fishes_/3843624).

Author's Response to Decision Letter for (RSOS-201033.R1)

See Appendix C.

Decision letter (RSOS-201033.R2)

Dear Dr Lu,

It is a pleasure to accept your manuscript entitled "3D segmentation of computed tomography data using Drishti Paint: new tools and developments" in its current form for publication in Royal Society Open Science.

on behalf of Professor Peter Haynes (Subject Editor)
openscience@royalsociety.org

Appendix A

I am pleased to provide a review of the contribution *A new tool for 3D segmentation of computed tomography data: Drishti Paint and its applications*. This paper provides an overview of the latest version of Drishti - an open source visualisation toolkit for slice based (e.g. CT data). Papers such as this are absolutely vital to raise awareness and foster the open source tomography community, and thus I believe this contribution is both important and timely - the only other publication documenting Drishti (and thus allowing citations of the software) is from a far earlier version. I have no doubt it will be well-cited, and provide an introduction for a wide range of researchers to Drishti, which is great.

I make a number of suggestions below, I would primarily identify as modifications to enhance the utility of this contribution to novices in the field. I'm aware they will involve more work from the authors (and apologise for this), and also that I may be trying to push the paper in a direction the authors did not intend. But I would encourage the authors to consider them carefully because they are there to help maximise the *value* of this paper to the community at large, and by acting on them the authors will reach a wider audience who will no doubt cite their work. The two main things I would highlight are:

-- I suggest an addition to the introduction to provide an overview of the landscape of open source packages out there, thus allowing the reader to know the options available to them and decide what software to use on the basis of their data, suited to their situation. This doesn't need to be long, but - especially given this is windows only - would be very valuable to readers.

-- When it comes to the workflow, I have highlighted many of the questions that popped into my head as I was reading. Papers such as this often follow a step by step approach:

<https://academic.oup.com/iob/article/2/1/obaa009/5818881#206006517>

And I recognise that this particular contribution is intended to show the possibilities drishti offers without this level of practical detail. However, I believe that if the authors were to provide some more detail regarding how to achieve the results they have (which are gorgeous) within the software (e.g. within the SI), then this would be really valuable for the reader. I am certain this would improve the impact of the paper, and provide a useful resource for the community for years to come.

I note also - primarily for Ajay - that I have tried, and failed, to compile the software on Ubuntu 18.04. I gave it half a day (I have lots of teaching to write for Autumn delivery, and this is the maximum amount of time I could set aside), at the end of which I was still getting a whole bunch of library errors. I appreciate the difficulty of release software on Linux, and so feel his pain, but wanted to highlight that for SPIERS, I managed to get the entire build process down to ~5 lines of bash using system packages:

<https://github.com/palaeoware/SPIERS>

It would be really valuable if the same approach were possible with Drishti, and no doubt allow more users to find and use the software. Just a thought.

In summary, I make a number of suggestions that I hope the authors will find beneficial, and ensure that this paper reaches the maximum number of people. I hope they are useful, and thank the authors for spending the time writing a contribution highlighting open source CT software for the community. I am happy to be identified as reviewer, and they should contact me if they have any questions.

-- Russell Garwood

russell.garwood@gmail.com

[I do not write anonymous reviews]

Abstract

-- Computational tomography → **Computed** tomography

--for its non-destructive and high-resolution in - **as a non-destructive and high-resolution means of**

-- which sheds light into → which sheds light **on**

-- However, how to efficiently and precisely reconstruct computed tomography data and better represent the data remains a hassle - this sentence doesn't mean much. It would be better replaced with something that says why people struggle to reconstruct data, or what hole this paper is trying to fill ("a hassle" can mean many things!)

-- and other advanced applications - this is a bit vague, can you be more specific? For example Drishti performs surfacing and volume rendering, it doesn't do multibody dynamics, FEA, CFD, or other "advanced applications"

-- computed tomography data thus benefit the scientific → computed tomography data **and** thus benefit the scientific

Introduction

-- that represent the wanted feature → that represent the **desired** feature

-- a more direct way for the reconstruction of 3D structures → a more direct way **to reconstruct** 3D structures; also, in what way is volume rendering more direct? You to surface you have to threshold, to volume render you have to create a transfer function. These are essentially the same thing, and this equally direct/indirect

-- However, the importance of scientific visualization software along with their functionality was underestimated by the community due to lack of public exposure and communications between multidiscipline. - I'm not sure this is true: have been in CT a while and run a CT lab, visualisation software is, in my experience, more important to most users of CT facilities than (say) data collection. There are many things you can say about the visualisation software landscape, so I suggest you replace this statement with something more specific about what need you're trying to fill with this paper (for example this is an open source segmentation solution that then allows you to volume render data - there are relatively few free solutions outside drishti that allow you to do this).

-- The 3D segmenting methods rely on thresholding, edge detection, clustering, or region growing to group pixels based on brightness, colour, or texture ← and then render them as discrete objects. I think it would be useful to highlight this point in this sentence.

-- Compared with researches carried out in the image processing field, more requirements have to be met towards building and developing applications which require the use of more efficient and effective techniques to save time and resources. - I don't think this is true. The amount of work, for example, that has gone into developing imagej to save time and resources is immense. And it's not a competition. This point does not add anything important to the paper, so I suggest you remove it.

-- and the last developed - most recently? New? I'm not quite sure what last developed means in this context

-- Here we present the latest ← this sentence is very long and has lots of brackets in it - I suggest you split it up so it is easier to follow (also, perhaps "the latest release of Drishti"

-- New features in Drishti, such as mesh generation and simplification, allow 3D printing, model simulation, etc.) are also covered in this study. ← this sentence needs to have an opening bracket somewhere

-- I suggest as part of this introduction the authors add a short paragraph as to how this fits into the range of options available for doing this kind of workflow, and where Drishti's strengths lie. Sticking to open source examples, Drishti and 3Dslicer are the only open source packages (of which I am aware) that do volume rendering, and the segmentation tools in Drishti are advanced (plus it makes beautiful renders!). But it does require datasets to be loaded into graphics memory if volume rendering, limiting the size of datasets that can be reconstructed on normal PCs (though I note that on the basis of this contribution it now does surfacing from paint). 3Dslicer and SPIERS both allow surfacing without the volume rendering stage, which may be better for some instances of really messy data, and SPIERS at least is designed to require mediocre hardware, so it may be preferable to readers without much computing power available if Drishti's visualiser struggles on their system. Drishti is only available on windows, whereas 3Dslicer and SPIERS are both available for Mac, and slicer is on Linux too. Surfacing allows meshes to be handled, which can then be modified in MeshLab (I see you mention this later) and Blender, if required. Slices can be handled in ImageJ, but this is not ideal for 3D work.

By adding this kind of information, it will allow the reader to make an informed decision about which software to learn, which I think would be very valuable, and help get this paper cited. Relevant papers include:

Sutton, M.D., Garwood, R.J., Siveter, D.J. and Siveter, D.J., 2012. SPIERS and VAXML; a software toolkit for tomographic visualisation and a format for virtual specimen interchange. *Palaeontologia Electronica*, 15(2), pp.1-14.

Dorador, J. and Rodríguez-Tovar, F.J., 2020. CroSSED sequence, a new tool for 3D processing in geosciences using the free software 3DSlicer. *Scientific Data*, 7(1), pp.1-8.

Garwood, R. and Dunlop, J., 2014. The walking dead: Blender as a tool for paleontologists with a case study on extinct arachnids. *Journal of Paleontology*, 88(4), pp.735-746.

Pye, F., Raja, N.B., Shirley, B., Kocsis, Á.T., Hohmann, N., Murdock, D.J. and Jarochowska, E., 2019. ImageJ and 3D Slicer: open source 2/3D morphometric software (No. e27998v1). *PeerJ Preprints*.

Materials and Methods

-- resolution of 21 μ m and 2 μ m respectively ← voxel size != spatial resolution. If this is the latter (i.e. the smallest thing you can tell apart, so at least three voxels), I suggest you explicitly state spatial resolution, if the former I suggest you say voxel size for clarity.

-- The whole specimen and the separated right anterior upper tooth plate were scanned in 2015 [27, 28] at CT Lab using a HeliScan MicroCT system ← scans for reference 27 were done on a custom built machine, not a heliscan, and so this citation is not appropriate here. According to 28 the scan parameters were: *A 2015 rescan used the new double-helix HeliScan CT Scanner with higher resolution and a new algorithm to process and reconstruct CT data. Both 1.2 mm aluminium and 0.35 mm stainless steel filters were used, with specimen distance 85 mm from the source, and detector position 396 mm from the source, and probed separately with a polychromatic X-ray beam (Bremsstrahlung radiation). Accelerating voltage of the electron beam generating the Bremsstrahlung radiation was 110 kV with a current of 100 μ A. A series of X ray transmission radiographs, collectively called the projection data, were acquired by the detector as the specimen was rotated through 360° double-helically over a period of 18 h. Reconstruction was based on 2520 radiographic projections formed on a 2048 × 1538 Varian Flat Panel camera.* I assume this is the same scan? If so I suggest either including this information here, or referring the reader to 28 for full scanning details.

-- CT data was reconstructed using an in-house software called Mango - using what kind of reconstruction algorithm? I don't think we need details, but it'd be good to know if this was an analytical (e.g. filtered back projection) v.s. An iterative reconstruction technique

-- The individual data sets were read at low resolution ← Why? This is an important practical detail and I think the reader will probably want to know why

-- incremented with the help of histograms ← how? Did you eyeball it, or did you have set windowing parameters you used?

-- slides were filtered ← slices, rather than slides, I assume. Which filters did you use and why? What advantages did this give, and what were the drawbacks?

Results

-- we also provide additional information about Drishti, such as installation instructions, a summary of all Drishti-supported import formats ← this is a really useful addendum, very nice!

-- Drishti Paint uses a combination of discontinuity detection based and similarity detection-based image segmentation approaches ← curves uses neither of these approaches, no? If so, I suggest rewording to include the manual nature of curves

-- Two transformations in mathematical morphology (i.e. erosion and dilation) were implemented in Drishti Paint a - how do these work? What effect do they have? When should the user use them? Why did you use them? What combination did you use, i.e. in which order? How did they speed up segmentation? Are there any potential drawbacks in doing so?

-- interest on the right anterior upper toothplate have been segmented from the original dataset of the toothplate (figure 3). - how? Using what tools?

-- Processed data then exported as separate volumes -- Why? What steps did you need to go through to do this?

-- Both segmentations were carried out in 16-bits full resolution - what advantages does working with 16 bit data give? And what disadvantages? What should a reader do if they want to follow your workflow with their fossil?

-- used type 2 gradient thresholding with values thresholding to select the range of the histogram for segmenting. - what steps did you follow to do this? When can this approach can be used? What happens in drishti if your structures of interest overlap in histogram space, but are not attached, which approach should we use to threshold them?

-- We suggest that these two transforms are more useful when using clean and high contrast dataset ← most fossil datasets are neither clean nor high contrast in my experience. What should I do in this case, use another piece of software, or another tool in Drishti?

-- as a risk of changing the morphology of the input datasets by doing multiple transformations at once - what impact could this have? How would we tell?

-- (i.e. use both values and gradient thresholding) - I think this needs unpacking/explaining a little bit

-- is also very sensible to noise and intensity → is also very **sensitive** to noise and intensity

-- specifically and choose carefully per dataset ← what would you advise a novice, new to drishti, should use to inform this choice?

-- also been extracted as surface meshes ← how? I assume this involves some kind of thresholding then a marching cubes to create the surface?

-- 3D models can then be used to generate - or do public outreach - e.g. Rahman, I.A., Adcock, K. and Garwood, R.J., 2012. Virtual fossils: a new resource for science communication in paleontology. *Evolution: Education and Outreach*, 5(4), pp.635-641. (forgive the self citations but I'm lazy and limited on time, so I know this is relevant!)

Appendix B

General Remarks

We thank both reviewers for very useful comments and suggestions. We accepted most of the comments and improved the manuscript accordingly.

Point-to-point Responses

Reviewer 1

"This manuscript presents a new version of the Drishti software, and a new tool for segmentation: Drishti Paint. I very much appreciate the fact that this software is open source, as it is always good to have free alternatives. However, I believe it would have been even more interesting to present and discuss the tools and features offered by this software for the segmentation process, which are not presented in details."

Response We thank the reviewer for the valuable comments. We have now added more detail regarding the new tools developed in version v2.7. Lines 154-170 is an entirely new section ("Gradient type 1 uses the magnitude of central difference (i.e. surface gradient vector). Gradient type 2 takes the sum of all voxel values in one neighbourhood (i.e. 3x3x3 box), then subtracts the central value from the above sum and divides it by 10. The absolute of the difference between the central value and the above result after dividing by 10, is clamped a value between 0.0 and 1.0. Gradient type 3 takes the sum of all voxel values in two neighbourhoods (i.e. 5x5x5 box), then subtracts the central value from the above sum and divides it by 70. As for type 2, the absolute of this difference is also clamped a value between 0.0 and 1.0. In gradient types 2 and 3 the 'gradient' value is shifted according to the voxel value (i.e. gradients for lower voxel values will be smaller compared to those for higher voxel values). For our case study, type 2 gradient thresholding combined with values thresholding were used to select the range of the histogram for segmenting out the right cheek complex. Gradient thresholding is also sensitive to noise and intensity inhomogeneities. There is no best combination when it comes to which gradient type one should use. Generally speaking, most boundaries can be clarified using gradient type 2 and 3. In contrast, gradient type 1 is more applicable with a cleaner dataset with fewer phases, such as medical datasets.") More detail on the segmentation process is flagged in the main text (lines 102-103: "see Supplementary Information for the detailed segmentation procedure."; lines 109-111: "Detailed information on new tools -gradient thresholding and 3D Freeform Painter is provided in the Supplementary Information.") The Supplementary Information has new text and figures added at lines 106-328.

"Actually, the segmentation process is only very vaguely discussed ("Two transformations in mathematical morphology (i.e. erosion and dilation) were implemented in Drishti Paint and we used a combination of these two transformations with 3D Freeform Painter to help with a faster segmentation process." l.107-109), and the data do not permit replication of the results because the raw 3D data are not provided. Thus the segmentation cannot be replicated."

Response We have now cropped the original CT data and made the raw data of right cheek complex available for readers to replicate our results. We have now removed the toothplate example as the original CT data is too large to share, and includes non-published material. This is explained in lines 98-101: "The raw data of the whole specimen was then cropped to focus on a selected region to include the raw data for right cheek complex, which was segmented out and is used as a case study here. Raw data of the selected region of interest can be downloaded from Figshare in the format *.pvl.nc (i.e. processed volume format)."

"Thus, I suggest to carefully check the English before publication, to provide the raw 3D data used for the segmentation, and to discuss further the detailed segmentation procedure and the Drishti features used."

Response We have asked Dr Gavin Young to help us with proof reading to improve the English, and there are numerous minor amendments and clarification throughout the text. (see tracked changes). We now provide the raw 3D data as supplementary data (available from the fig share repository: <https://figshare.com/s/244e5f407d331a39b402>). We have added further discussion of

the segmenting procedure. [see lines 100-103:“We developed a general protocol to segment the 3D volumetric data using Drishti Paint v2.7. (see Supplementary Information for the detailed segmentation procedure).”; lines 132-136:“Dilation adds pixels to the boundaries of objects in an image, while erosion removes pixels on object boundaries. Morphological dilation makes objects more visible and fills in small holes in objects. Morphological erosion removes islands and small objects so that only substantive objects remain. These two transformations can be used in any order.”; lines 140-145: “ We recommended keeping a copy of the original volume data, before doing multiple transformations, to avoid losing any information of the original dataset. Interpolation, is another tool which can be used in Curve mode. However, interpolation is not recommended for palaeontological datasets (due to lack of reproducibility), so is not applicable for our case study, and is not considered further in this paper.”; lines 154-170:“Gradient type 1 uses the magnitude of central difference (i.e. surface gradient vector). Gradient type 2 takes the sum of all voxel values in one neighbourhood (i.e. 3x3x3 box), then subtracts the central value from the above sum and divides it by 10. The absolute of the difference between the central value and the above result after dividing by 10, is clamped a value between 0.0 and 1.0. Gradient type 3 takes the sum of all voxel values in two neighbourhoods (i.e. 5x5x5 box), then subtracts the central value from the above sum and divides it by 70. As for type 2, the absolute of this difference is also clamped a value between 0.0 and 1.0. In gradient types 2 and 3 the ‘gradient’ value is shifted according to the voxel value (i.e. gradients for lower voxel values will be smaller compared to those for higher voxel values). For our case study, type 2 gradient thresholding combined with values thresholding were used to select the range of the histogram for segmenting out the right cheek complex. Gradient thresholding is also sensitive to noise and intensity inhomogeneities. There is no best combination when it comes to which gradient type one should use. Generally speaking, most boundaries can be clarified using gradient type 2 and 3. In contrast, gradient type 1 is more applicable with a cleaner dataset with fewer phases, such as medical datasets.” and Supplementary Information lines 106-328].

“I suggest an addition to the introduction to provide an overview of the landscape of open source packages out there, thus allowing the reader to know the options available to them and decide what software to use on the basis of their data, suited to their situation.”

Response We accept this, and have added a summary [see lines 42-45:“ A range of well established commercial software, such as Mimics, VG Studio and AVIZO, which provide numerous functionalities and good rendering outputs. However, the potential of open-source software, which is both freely available and easy to access, has not been fully explored.”; lines 47-49:“2D segmentation uses each image or slice in a volumetric dataset to construct a 3D presentation.”; lines 52-59: “Amongst ten well-known cross-platform software programs [23], five [23] use both surface and volume rendering, but only Drishti [24] with ‘an intuitive user interface’ [23], can perform 3D segmentation directly on a volume (using the new tool presented here, 3D Freeform Painter). Drishti uses direct volume rendering with voxel ray casting and texture slicing algorithms; combinations of local and global illuminations along with a 2D transfer function which merges colour and opacity to volume to provide realism on different materials and textures through lighting, shading, or shadowing via different user-generated light volumes and change of opacity.”].

“When it comes to the workflow, I have highlighted many of the questions that popped into my head as I was reading. Papers such as this often follow a step by step approach: <https://academic.oup.com/iob/article/2/1/obaa009/5818881#206006517>. And I recognise that this particular contribution is intended to show the possibilities drishti offers without this level of practical detail. However, I believe that if the authors were to provide some more detail regarding how to achieve the results they have (which are gorgeous) within the software (e.g. within the SI), then this would be really valuable for the reader. I am certain this would improve the impact of the paper, and provide a useful resource for the community for years to come.”

Response We thank the reviewer for very positive comments. We have now provided more detail regarding how we achieved our “gorgeous” results in lines 123-130 (“The right cheek complex has been segmented from the original CT dataset for our case study. Figure 2 shows an overview of our

case study, where a fossil fish skull (Fig. 2a-b) is cropped (Fig. 2c) and segmented using the latest tools in Drishti v2.7 (Fig. 2d-e). Processed data is exported as a separate volume by using the tagging function to extract the region of interest. Segmentation (in this cheek complex example) was carried out in 16-bits full resolution in alignment with the raw data to include all information from the original scan (8-bits resolution could be used, if computing power is limited, as noted above.”), and lines 106-387 (Supplementary Information).

“I note also - primarily for Ajay - that I have tried, and failed, to compile the software on Ubuntu 18.04. I gave it half a day (I have lots of teaching to write for Autumn delivery, and this is the maximum amount of time I could set aside), at the end of which I was still getting a whole bunch of library errors. I appreciate the difficulty of release software on Linux, and so feel his pain, but wanted to highlight that for SPIERS, I managed to get the entire build process down to ~5 lines of bash using system packages:<https://github.com/palaeoware/SPIERS>. It would be really valuable if the same approach were possible with Drishti, and no doubt allow more users to find and use the software. Just a thought.”

Response We have sent out the Linux version of Drishti to Reviewer #1 for feedback however this major task requires quite amount of time, effort and sufficient testing before the final launch. As a result, We will provide the Linux version to users on the GitHub site once it has been fully tested [see lines 205-206]. We have acknowledged Russell Garwood’s contribution towards testing the Linux version of Drishti v2.7 [see lines 227-228].

All minor amendments of Review 1 have been corrected: computed tomography (line 14); as a non-destructive and high resolution means of [line 15]; sheds light on [line 17]; computed tomography data and thus benefit the scientific [we have deleted this sentence]; that represent the desired feature [line 39]; is also very sensitive to noise and intensity [line 166].

Abstract

-- However, how to efficiently and precisely reconstruct computed tomography data and better represent the data remains a hassle - this sentence doesn’t mean much. It would be better replaced with something that says why people struggle to reconstruct data, or what hole this paper is trying to fill (“a hassle” can mean many things!)

Response We have changed this sentence [lines 17-19]: “3D segmentation of CT data is supported by various well established software programs, but the powerful functionalities and capabilities of open-sourced software have not been fully revealed.”

-- and other advanced applications - this is a bit vague, can you be more specific? For example Drishti performs surfacing and volume rendering, it doesn’t do multibody dynamics, FEA, CFD, or other “advanced applications”

Response Accepted. We have expanded “and other advanced applications” to be more specific [lines 19-24]: “Here, we present a new release of the open-source volume exploration, rendering, and 3D segmentation software, *Drishti* v2.7. We introduce a new tool for thresholding volume data (i.e. gradient thresholding), and a protocol for performing 3D segmentation using the 3D Freeform Painter tool. These new tools and workflow enabling more accurate and precise digital reconstruction, 3D modelling and 3D printing results.”

Introduction

-- a more direct way for the reconstruction of 3D structures → a more direct way to reconstruct 3D structures; also, in what way is volume rendering more direct? You to surface you have to threshold, to volume render you have to create a transfer function. These are essentially the same thing, and this equally direct/indirect

Response We have revised this [see lines 38-40]: “surface rendering, the method of interpreting datasets by generating a set of polygons that represent the desired feature; and volume rendering [10], for the reconstruction of 3D structures both internally and externally”.

-- However, the importance of scientific visualization software along with their functionality was underestimated by the community due to lack of public exposure and communications between multidiscipline. - I’m not sure this is true: have been in CT a while and run a CT lab, visualisation software is, in my experience, more important to most users of CT facilities than (say) data collection. There are many things you can say about the visualisation software landscape, so I suggest you replace this statement with something more specific about what need you’re trying to fill with this paper (for example this is an open source segmentation solution that then allows you to volume render data - there are relatively few free solutions outside drishti that allow you to do this).

Response Accepted. As noted above we have changed and expanded this text. See lines 42-59: “A range of well established commercial software, such as Mimics, VG Studio and AVIZO, which provide numerous functionalities and good rendering outputs. However, the potential of open-source software, which is both freely available and easy to access, has not been fully explored. 3D segmentation, segmenting the internal region of interest in sequences of images, is a vital tool for investigating and understanding the internal structures of target objects [6, 20-22]. 2D segmentation uses each image or slice in a volumetric dataset to construct a 3D presentation. In contrast, 3D segmenting methods, the main focus of this paper, using thresholding, edge detection, clustering, or region growing techniques to group pixels based on brightness, colour, or texture [21] and then render them as discrete objects. Amongst ten well-known cross-platform software programs [23], five [23] use both surface and volume rendering, but only Drishti [24] with ‘an intuitive user interface’ [23], can perform 3D segmentation directly on a volume (using the new tool presented here, 3D Freeform Painter). Drishti uses direct volume rendering with voxel ray casting and texture slicing algorithms; combinations of local and global illuminations along with a 2D transfer function which merges colour and opacity to volume to provide realism on different materials and textures through lighting, shading, or shadowing via different user-generated light volumes and change of opacity.”

-- The 3D segmenting methods rely on thresholding, edge detection, clustering, or region growing to group pixels based on brightness, colour, or texture ← and then render them as discrete objects. I think it would be useful to highlight this point in this sentence.

Response Agreed, and we have added “and then render them as discrete objects” to line 51.

-- Compared with researches carried out in the image processing field, more requirements have to be met towards building and developing applications which require the use of more efficient and effective techniques to save time and resources. - I don’t think this is true. The amount of work, for example, that has gone into developing imagej to save time and resources is immense. And it’s not a competition. This point does not add anything important to the paper, so I suggest you remove it.

Response Accepted, we have now removed this sentence.

-- and the last developed - most recently? New? I’m not quite sure what last developed means in this context

Response We have clarified this and removed “the last developed” as suggested [line 62].

-- Here we present the latest ← this sentence is very long and has lots of brackets in it - I suggest you split it

up so it is easier to follow (also, perhaps “the latest release of Drishti”)

Response Accepted, we have split this sentence to make it easier to follow. See lines 62-67: “Here we present the most recent release of Drishti v2.7, and introduce new tools for 3D segmentation (2D and 3D painters) of volumetric data. We also introduce a new tool- gradient threshold, and suggest protocols for how to perform 3D segmentation using Drishti Paint v2.7 efficiently and precisely, using the CT scan data of a fossil fish as a case study. New features in Drishti, such as mesh generation and simplification, producing 3D surface models are also explained and discussed.”

-- New features in Drishti, such as mesh generation and simplification, allow 3D printing, model simulation, etc.) are also covered in this study. ← this sentence needs to have an opening bracket somewhere

Response Brackets removed in line 67.

-- I suggest as part of this introduction the authors add a short paragraph as to how this fits into the range of options available for doing this kind of workflow, and where Drishti’s strengths lie. Sticking to open source examples, Drishti and 3DSlicer are the only open source packages (of which I am aware) that do volume rendering, and the segmentation tools in Drishti are advanced (plus it makes beautiful renders!). But it does require datasets to be loaded into graphics memory if volume rendering, limiting the size of datasets that can be reconstructed on normal PCs (though I note that on the basis of this contribution it now does surfacing from paint). 3DSlicer and SPIERS both allow surfacing without the volume rendering stage, which may be better for some instances of really messy data, and SPIERS at least is designed to require mediocre hardware, so it may be preferable to readers without much computing power available if Drishti’s visualiser struggles on their system. Drishti is only available on windows, whereas 3DSlicer and SPIERS are both available for Mac, and slicer is on Linux too. Surfacing allows meshes to be handled, which can then be modified in MeshLab (I see you mention this later) and Blender, if required. Slices can be handled in ImageJ, but this is not ideal for 3D work. By adding this kind of information, it will allow the reader to make an informed decision about which software to learn, which I think would be very valuable, and help get this paper cited. Relevant papers include:

Sutton, M.D., Garwood, R.J., Siveter, D.J. and Siveter, D.J., 2012. SPIERS and VAXML; a software toolkit for tomographic visualisation and a format for virtual specimen interchange. *Palaeontologia Electronica*, 15(2), pp.1-14.

Dorador, J. and Rodríguez-Tovar, F.J., 2020. CroSSED sequence, a new tool for 3D processing in geosciences using the free software 3DSlicer. *Scientific Data*, 7(1), pp.1-8.

Garwood, R. and Dunlop, J., 2014. The walking dead: Blender as a tool for paleontologists with a case study on extinct arachnids. *Journal of Paleontology*, 88(4), pp.735-746.

Pye, F., Raja, N.B., Shirley, B., Kocsis, Á.T., Hohmann, N., Murdock, D.J. and Jarochowska, E., 2019. ImageJ and 3D Slicer: open source 2/3D morphometric software (No. e27998v1). *PeerJ Preprints*.

Response Accepted. We have added all suggested references and revised our introduction accordingly. See lines 68-75: “Volume rendering does require datasets to be loaded into graphics memory. The minimum requirements to run Drishti are: graphics processing unit (GPU)- Nvidia GT 630 with 1GB memory and random access memory (RAM)- 4GB. Drishti does not have a specific requirement for the central processing unit (CPU). For readers who do not have much computing power, we suggest trying other open-source software such as SPIERS [28] and 3D slicer [29-30]. Both software perform excellent surface rendering and allow meshes to be handled, which can then be modified in Meshlab [31] or Blender [32].”

Materials and Methods

-- resolution of 21 μ m and 2 μ m respectively ← voxel size != spatial resolution. If this is the latter (i.e. the smallest thing you can tell apart, so at least three voxels), I suggest you explicitly state spatial resolution, if the former I suggest you say voxel size for clarity.

Response Accepted. We have now changed “resolution” to “voxel size” [see line 83].

-- The whole specimen and the separated right anterior upper tooth plate were scanned in 2015 [27, 28] at CT Lab using a HeliScan MicroCT system ← scans for reference 27 were done on a custom built machine, not a heliscan, and so this citation is not appropriate here. ... I assume this is the same scan? If so I suggest either including this information here, or referring the reader to 28 for full scanning details.

Response As noted above, we have removed the scanning information for the upper toothplate, and we now refer readers to reference number 33 for full CT scanning information [see line 83].

-- CT data was reconstructed using an in-house software called Mango - using what kind of reconstruction algorithm? I don't think we need details, but it'd be good to know if this was an analytical (e.g. filtered back projection) v.s. An iterative reconstruction technique

Response We have now added required information on *Mango* as provided by Department of Applied Maths Colleague Dr Andrew Kingston, one of the developer of *Mango*. See lines 86-95: "Dr A. Kingston (ANU Department of Applied Mathematics) has provided the following summary [pers.comm., 8 Sep 2020]: Mango can perform 3D tomographic reconstruction on CPU or GPU clusters. The CPU-only code is limited to analytic methods: Feldkamp/Davis/Kress (FDK) filtered back-projection (FBP) for a circular X-ray source trajectory and Katsevich FBP for helical/double-helical X-ray source trajectories [34]. The GPU code can also perform multigrid iterative reconstruction [35] for a space-filling X-ray source trajectory [1]. The reconstruction code has software for automatic geometric alignment capabilities [36], X-ray source movement correction [37], and component or rigid-body sample movement correction [38]."

-- The individual data sets were read at low resolution ← Why? This is an important practical detail and I think the reader will probably want to know why

Response Data in Drishti is read as the low-resolution mode by default but the user can press F2 to bring up the high-resolution mode. This enables users to overview their data in low-res mode and decide whether they want to proceed with rendering. If yes, then they can switch to the high-resolution mode, which takes more graphics memory.

-- incremented with the help of histograms ← how? Did you eyeball it, or did you have set windowing parameters you used?

Response We have now removed this sentence as data cropping and image processing are not the main focus of this paper (i.e. 3D segmentation). Readers can learn or read more information regards to these aspects on GitHub site and YouTube channel.

-- slides were filtered ← slices, rather than slides, I assume. Which filters did you use and why? What advantages did this give, and what were the drawbacks?

Response We have demonstrated the detailed segmentation procedure in Supplementary Information (lines 106-328), which is the main focus of this paper. We have removed sentences relating to data preparation (see tracked changes), as image processing is not considered here.

Results

-- we also provide additional information about Drishti, such as installation instructions, a summary of all Drishti-supported import formats ← this is a really useful addendum, very nice!
-- Drishti Paint uses a combination of discontinuity detection based and similarity detection-based image segmentation approaches ← curves uses neither of these approaches, no? If so, I suggest rewording to include the manual nature of curve

Response We thank the reviewer, and reworded our sentences to clarify. See lines 119-122: “Drishti Paint uses a variety of discontinuity detection-based and similarity detection-based image segmentation approaches. These two approaches are both implemented in two modes in Drishti Paint v2.7, i.e. ‘Graph Cut’ and ‘Curve’ (previously known as livewire).”

-- Two transformations in mathematical morphology (i.e. erosion and dilation) were implemented in Drishti Paint a - how do these work? What effect do they have? When should the user use them? Why did you use them? What combination did you use, i.e. in which order? How did they speed up segmentation? Are there any potential drawbacks in doing so?

Response We have now included a simple introduction for the two transformations [see lines 132-136]: “Dilation adds pixels to the boundaries of objects in an image, while erosion removes pixels on object boundaries. Morphological dilation makes objects more visible and fills in small holes in objects. Morphological erosion removes islands and small objects so that only substantive objects remain. These two transformations can be used in any order.”

-- interest on the right anterior upper toothplate have been segmented from the original dataset of the toothplate (figure 3). - how? Using what tools?

Response As noted above, we have now removed the toothplate example as it is not published and the original dataset is too large for sharing.

-- Processed data then exported as separate volumes -- Why? What steps did you need to go through to do this?

Response Accepted, and we now explain how to export volume in the Supplementary Information [lines 241-265].

-- Both segmentations were carried out in 16-bits full resolution - what advantages does working with 16 bit data give? And what disadvantages? What should a reader do if they want to follow your workflow with their fossil?

Response 16-bits full resolution of the raw data has the advantage of full information of the scanned data. However, users can then sub-sample their data to a lower resolution to follow our workflow. We have clarified this point [lines 126-130]: “ Processed data is exported as a separate volume by using the tagging function to extract the region of interest. Segmentation (in this cheek complex example) was carried out in 16-bits full resolution in alignment with the raw data to include all information from the original scan (8-bits resolution could be used, if computing power is limited, as noted above).”

-- used type 2 gradient thresholding with values thresholding to select the range of the histogram for segmenting.
- what steps did you follow to do this? When can this approach can be used? What happens in drishti if your

structures of interest overlap in histogram space, but are not attached, which approach should we use to threshold them?

Response We have now explained the three different types of gradient thresholds and their suggested usage by adding new text [see lines 151-170]: “Multiple-thresholding (i.e. use both values and gradient thresholding) in Drishti Paint v2.7 is beneficial for volume segmentation and usually is the first step towards segmenting a volume (Figure 2d). Gradient type 1 uses the magnitude of central difference (i.e. surface gradient vector). Gradient type 2 takes the sum of all voxel values in one neighbourhood (i.e. 3x3x3 box), then subtracts the central value from the above sum and divides it by 10. The absolute of the difference between the central value and the above result after dividing by 10, is clamped a value between 0.0 and 1.0. Gradient type 3 takes the sum of all voxel values in two neighbourhoods (i.e. 5x5x5 box), then subtracts the central value from the above sum and divides it by 70. As for type 2, the absolute of this difference is also clamped a value between 0.0 and 1.0. In gradient types 2 and 3 the ‘gradient’ value is shifted according to the voxel value (i.e. gradients for lower voxel values will be smaller compared to those for higher voxel values). For our case study, type 2 gradient thresholding combined with values thresholding were used to select the range of the histogram for segmenting out the right cheek complex. Gradient thresholding is also sensitive to noise and intensity inhomogeneities. There is no best combination when it comes to which gradient type one should use. Generally speaking, most boundaries can be clarified using gradient type 2 and 3. In contrast, gradient type 1 is more applicable with a cleaner dataset with fewer phases, such as medical datasets.”

-- We suggest that these two transforms are more useful when using clean and high contrast dataset ← most fossil datasets are neither clean nor high contrast in my experience. What should I do in this case, use another piece of software, or another tool in Drishti?

Response We have now added a clarification after this sentence [see lines 140-142]: “We recommended keeping a copy of the original volume data, before doing multiple transformations, to avoid losing any information of the original dataset.”

-- as a risk of changing the morphology of the input datasets by doing multiple transformations at once - what impact could this have? How would we tell?

Response The impact after multiple transformations depends on the dataset. The only way to tell the differences is to save the original data then proceed with all transformations in an identical copy of the original, so the differences can be assessed.

-- (i.e. use both values and gradient thresholding) - I think this needs unpacking/explaining a little bit

Response Accepted. We have added new text to explain this in lines 146-170:

“ Three types of gradient thresholding are developed and implemented in Drishti Paint v2.7. Values thresholding was already developed, but this is the first implementation on gradient thresholding in an open-source volume rendering software program. Gradient thresholding can be combined with values thresholding to clarify and more precisely identify the boundaries between different phases, which then makes 3D segmentation process easier. Multiple-thresholding (i.e. use both values and gradient thresholding) in Drishti Paint v2.7 is beneficial for volume segmentation and usually is the first step towards segmenting a volume (Figure 2d).

Gradient type 1 uses the magnitude of central difference (i.e. surface gradient vector). Gradient type 2 takes the sum of all voxel values in one neighbourhood (i.e. 3x3x3 box), then subtracts the central value from the above sum and divides it by 10. The absolute of the difference between the central value and the above result after dividing by 10, is clamped a value between 0.0 and 1.0. Gradient type 3 takes the sum of all voxel values in two neighbourhoods (i.e. 5x5x5 box), then subtracts the central value from the above sum and divides it by 70. As for type 2, the absolute of this difference is also

clamped a value between 0.0 and 1.0. In gradient types 2 and 3 the 'gradient' value is shifted according to the voxel value (i.e. gradients for lower voxel values will be smaller compared to those for higher voxel values). For our case study, type 2 gradient thresholding combined with values thresholding were used to select the range of the histogram for segmenting out the right cheek complex. Gradient thresholding is also sensitive to noise and intensity inhomogeneities. There is no best combination when it comes to which gradient type one should use. Generally speaking, most boundaries can be clarified using gradient type 2 and 3. In contrast, gradient type 1 is more applicable with a cleaner dataset with fewer phases, such as medical datasets. "

-- specifically and choose carefully per dataset ← what would you advise a novice, new to drishti, should use to inform this choice?

Response We have now added a general suggestion [see lines 149-153]: "Gradient thresholding can be combined with values thresholding to clarify and more precisely identify the boundaries between different phases, which then makes 3D segmentation process easier. Multiple-thresholding (i.e. use both values and gradient thresholding) in Drishti Paint v2.7 is beneficial for volume segmentation and usually is the first step towards segmenting a volume (Figure 2d)."

-- also been extracted as surface meshes ← how? I assume this involves some kind of thresholding then a marching cubes to create the surface?

Response We have now added more information related to the procedure of extracting surface mesh in the Supplementary Information [lines 267-363].

-- 3D models can then be used to generate - or do public outreach - e.g. Rahman, I.A., Adcock, K. and Garwood, R.J., 2012. Virtual fossils: a new resource for science communication in paleontology. *Evolution: Education and Outreach*, 5(4), pp.635-641. (forgive the self citations but I'm lazy and limited on time, so I know this is relevant!)

Response This is a very relevant reference, and has been added [see line 189].

Reviewer 2

“...A protocol would describe in detail the procedure used in Drishti Paint to segment the elements from the original scan data. The original scan data is the raw data that should be included for users to replicate the results from, not the resulting segmented pvl.nc file. I cannot segment out something that has already been segmented. I understand that the authors may not want to release a microCT dataset of a specimen that has not been described yet but if that is so they should use a microCT dataset that they are willing to make freely available. They could even download one from elsewhere that has been published.”

Response Accepted. We now include a detailed segmentation procedure using the right cheek complex in the Supplementary Information, and we have cropped the original data around the region of interest which is now uploaded to Figshare. As noted above, we have removed the toothplate data from this paper, as it contains non-published material.

“Furthermore, the authors do not describe anything new from that which is freely available in the instructions and tutorials on the Drishti github page. In fact, the material available there is far more informative in terms of segmentation protocols.”

Response The purpose of this paper is not teaching readers how to use a particular function/features in Drishti (unlike YouTube tutorials which teach one thing at a time, or the GitHub site which includes general information). Our original aim is not making this manuscript look like a manual or instruction. Much work has already gone into YouTube and GitHub, and these are generally available and can be referred to by any user. The aim of this paper (as Reviewer #1 pointed out) is to let the community know they have another option when they need software for segmenting volumetric data. We would encourage readers to go online and explore themselves rather than setting a path for them (one publication can not cover everything and a given workflow will not suit all datasets). However, as suggested by both reviewers, we now include more detailed segmentation instructions in the Supplementary Information [lines 106-328].

“Make the entire microCT dataset available. If the authors do not wish to make this dataset available then use a different one. If they do not have a different dataset to use then they could crop the original dataset to around the region of interest to provide enough data that the user could segment the cheek and tooth but not be able to access all the other elements. This would solve the risk of another party describing the specimen before the authors do. It is the process this manuscript purports to describe that is important as this manuscript describes a method and not a specimen.” “From the website of Royal Society Open Science, one of the distinguishing features of the journal is that “articles embody open data principles”. Unless the original microCT scan data is released this manuscript does not adhere to the principles of the journal.” “Describe in detail each step that a user must do to in Drishti Paint to segment the specific element used in this example. Statements such as line 110-111 “The right cheek complex has been segmented from the original CT dataset of the whole specimen (figure 2).” does not tell me how to do this. These instructions could be included in the supplementary info.”

Response Accepted. As noted above, we have now cropped the original data around the region of interest and provide this data along with the segmentation procedure. [See Supplementary Information.]

“...In the case of the segmented placoderm tooth with the internal canals, the authors may have achieved better results using the ‘tubes’ function in Drishti Paint. Perhaps they did use this function or perhaps they used the gradient function, they do not say how they segmented the tubes. In fact, how do you access the tubes function in Drishti as shown in the online tutorial video?!? Spacebar to access the command help menu does not function in v2.6.6 or 2.7 in the windows zip files from github.”

Response Toothplate data is no longer included (see above). Drishti was being updated during preparation of our manuscript. We have now changed v2.6.6 to v2.7 throughout. Spacebar should be functional.

“Provide detailed compilation and installation instructions for linux and mac. The authors provided a further brief set of steps on how to install Drishti for linux following my request for information on this matter. It did not work. I understand that linux distributions are all different and an individual’s linux set up unlikely matches that of another but I was still unable to compile and install it after editing the .pri file for my setup and then subsequently altering the makefile produced by QTCreator to account for the versions of the dependencies that the authors provided me. Both systems are unstable and crash frequently in Wine (windows compatibility layer for linux operating systems).”

Response The reviewer previously requested information on Linux, and this was provided. The Linux system is still under development and testing which is a major task (there are many open-source programs only support 1-2 operation systems). At this stage, it is premature to release the Linux version. The same applies for Mac (up to v2.6.4 did support Mac), but we are a small team so Mac version of latest Drishti is a long term aim.

“Correct English grammar. Despite native English speakers being acknowledged for their assistance in proof-reading the English grammar is still not acceptable for publication.”

Response We have asked Dr Gavin Young to help with proof reading to improve the English.

Responses to “Other detailed points”

Title – Drishti Paint is not a new tool and has been present in many versions of Drishti

Response Accepted. Title is now revised.

Line 33 Should you include microCT and synchrotron data here as well as Drishti can handle all these. Yes, they are all essentially CT but perhaps not everyone knows that.

Response We have clarified this and changed “CT” to “CT, Micro-CT and synchrotron radiation phase-contrast imaging”[see line 33]

Lines 36-38 “For the last four decades, the field of 3D scientific visualization field has have been developed considerably with many new techniques available ways to visualize and analyse data more accurately [10-19].”

Response This sentence is now corrected. See lines 35-37: “For over three decades, the field of 3D scientific visualisation has been developed considerably with many new techniques to visualise and analyse data more accurately [10-19].”

Line 44. By which community? All communities, the palaeo, biological, medical communities?

Response This sentence has been removed and text revised as suggested by Reviewer #1. See lines 42-55: “A range of well established commercial software, such as Mimics, VG Studio and AVIZO, which provide numerous functionalities and good rendering outputs. However, the potential of open-source software, which is both freely available and easy to access, has not been fully explored. 3D segmentation, segmenting the internal region of interest in sequences of images, is a vital tool for investigating and understanding the internal structures of target objects [6, 20-22]. 2D segmentation uses each image or slice in a volumetric dataset to construct a 3D presentation. In contrast, 3D segmenting methods, the main focus of this paper, using thresholding, edge detection, clustering, or

region growing techniques to group pixels based on brightness, colour, or texture [21] and then render them as discrete objects. Amongst ten well-known cross-platform software programs [23], five [23] use both surface and volume rendering, but only Drishti [24] with ‘an intuitive user interface’ [23], can perform 3D segmentation directly on a volume (using the new tool presented here, 3D Freeform Painter).”

Line 46. 3D segmentation is not necessarily about segmenting structure but regions of interest. I would use ‘regions of interest’ rather than structures. It depends what type of data you are working with.

Response Accepted, “structure” changed to “the region of interest” [see line 46]

Lines 48 -50. You omit manual segmentation involving drawing the shape of each region of interest and, sometimes, using interpolation. Though perhaps the most commonly used in palaeo datasets it is also the most fraught with uncertainty, error and lack of reproducibility. This should be mentioned somewhere in the manuscript as you are presenting a workflow that is reproducible.

Response We agree that interpolation is not a great tool for paleontological datasets and this should be mentioned. We have now added two sentences mentioning interpolation [lines 142-145]: “ Interpolation, is another tool which can be used in Curve mode. However, interpolation is not recommended for palaeontological datasets (due to lack of reproducibility), so is not applicable for our case study, and is not considered further in this paper.”

Lines 50–52. This is not great English I’m afraid. I would reword along the lines of: “Compared with researches carried out in the image processing field, more requirements have to be met towards building and Despite this researchers in many fields still require the development of developing applications and techniques which are require the use of more efficient and effective techniques at segmenting to save research time and resources.”

Response We have revised the text and this sentence has been removed as suggested by Reviewer #1.

Line 54. “... high quality 3D rendering” better than “... superb visualisation outcomes.”. Other rendering packages are good as well so it might be worth explaining what gives Drishti rendering its unique quality. Ajay should be able to add this in.

Response We have now explained ‘what gives Drishti rendering its unique quality’ in some new text [lines 55-59]: “ *Drishti* uses direct volume rendering with voxel ray casting and texture slicing algorithms; combinations of local and global illuminations along with a 2D transfer function which merges colour and opacity to volume to provide realism on different materials and textures through lighting, shading, or shadowing via different user-generated light volumes and change of opacity.”

Line 55. Drishti 2.7 is now available. I assume that the same Drishti paint tools are present in 2.7 in which case the latest version should be cited.

Response *Drishti* was being updated as part of the preparation of our manuscript. We have now updated the version number and other aspects.

Lines 80-84. This is the crux (most important part) of the manuscript and it is treated very lightly. In the abstract the authors state they introduce a “...protocol for performing 3D segmentation” but they have not. They have simply stated that the data was segmented using Drishti Paint.

Response We have accepted all suggestions for both reviewers in regards to adding additional information for 3D segmentation and mesh generation procedures. This is now clarified in the Supplementary Information. [See lines 106-387.]

Appendix C

Details of changes requested by Reviewer 2

General typographical and grammatical comments of Reviewer 2 have been corrected:

- The word “which” has been removed [line 18];
- Changed “enabling” to “enable” [line 31];
- Changed “using” to “use” [line 68]
- Changed “a new toolset” to “new tools” [line 95]
- Removed “Nvidia GT 630” [line 130]
- Sentence deleted [lines 138-139]
- Added “was saved as 16 bit images” [line 148]
- Added a few words to define what filtering slices means- “slices were filtered by selecting the best range on the tomogram.” [lines 150-151]
- Added “the” before “right cheek complex” [line 152]
- Figshare link has been added to lines 153 & 398
- Deleted lines 156-160 as suggested by review 2
- Added “the” [line 179]
- Removed “...so is not applicable for our case study and is not considered further in this paper.” [lines 219-220]
- Added “the” [line 226]
- Added the suggested preface sentence “In the gradient thresholding tool..” [line 230]
- Changed “clamped” to a more general word “restricted” [lines 234 & 238]
- Full stop removed. [line 253]
- Inserted “see supplementary information” after “simplification” [lines 253-254]
- Changed “less information is resulting” to “the loss of information results” [line 258]
- Changed “scientific community” to “public” [line 353]
- Revised to “Supplementary Table” [Supplementary Information, line 86]

Notes on supplementary information

The following comment on how to avoid this error message added to Supplementary Information [page 14]: “When dilating the tagged region of interest, a pop-up message may appear- “cannot dilate. You are on voxel with tag 0, was expecting tag 1”. This occurs only when the cursor is not on a voxel with the appropriate tag value. The Near Neighbour switch under 3D Preview parameter panel gives a better view of voxels (default is trilinear interpolation of voxels which gives smoother rendering but may obstruct appropriate voxel selection). This helps to select the correct voxel.”